# Manganese Dioxide Nanoparticles Prepared by Laser Ablation as Materials with Interesting Electronic, Electrochemical, and Disinfecting Properties in Both Colloidal Suspensions and Deposited on Fluorine-Doped Tin Oxide

**DOI:** 10.3390/nano12224061

**Published:** 2022-11-18

**Authors:** Jhonatan Corrales, Jorge Acosta, Sandra Castro, Henry Riascos, Efraim Serna-Galvis, Ricardo A. Torres-Palma, Yenny Ávila-Torres

**Affiliations:** 1Maester in Chemical Sciencies, Faculty of technology, Universidad Tecnológica de Pereira, Pereira 660003, Colombia; 2Department of Macromolecular Compounds, Faculty of Chemistry, Lomonosov Moscow State University MSU, 119991 Moscow, Russia; 3Grupo de Investigación en Elctroquímica y Medio Ambiente, Universidad Santiago de Cali, Faculty of Sciences, Santiago de Cali 760035, Colombia; 4Grupo de Ablación Láser, Universidad Tecnológica de Pereira, Pereira 660001, Colombia; 5Grupo de Investigación Catalizadores y Adsorbentes (Catalad), Faculty of Exact and Natural Sciences, Chemistry Institution, Universidad de Antioquia UdeA, Calle 70 No. 52-21, Medellín 050010, Colombia; 6Grupo de Investigación en Remediación Ambiental y Biocatálisis (GIRAB), Faculty of Exact and Natural Sciences, Chemistry Institution, Universidad de Antioquia UdeA, Calle 70 No. 52-21, Medellín 050010, Colombia

**Keywords:** antimicrobial activity, α-MnO_2_ nanoparticles, deposited nanoparticles, photocatalysis, reactive oxygen species, fluorine-doped tin oxide

## Abstract

Nanoparticles (NPs) of α-MnO_2_ have high applicability in photoelectrochemical, heterogeneous photocatalysis, optical switching, and disinfection processes. To widen this panorama about MnO_2_ NPs, the formation of this material by laser ablation and deposition by dip-coating on fluorine-doped tin oxide (FTO), were considered in this study. The optical, spectroscopic, electrochemical characterization, and the evaluation of the antimicrobial activity, plus the photocatalytic response, were measured herein in colloidal media and deposited. For the deposition of NPs on FTO sheet, an anode is produced with a pseudocapacitive behavior, and 2.82 eV of band gap (GAP) in comparison with colloidal NPs for a value of 3.84 eV. Both colloidal suspension and deposited NPs have intrinsic antibacterial activity against two representative microorganisms (*E. coli* and *S. aureus*), and this biological activity was significantly enhanced in the presence of UVA light, indicating photocatalytic activity of the material. Thus, both the colloidal suspension and deposited NPs can act as disinfecting agents themselves or via light activation. However, an antibacterial behavior different for *E. coli* and *S. aureus* was observed, in function of the aggregation state, obtaining total *E. coli* disinfection at 30 min for deposited samples on FTO.

## 1. Introduction

Nanometer-size semiconductors are gaining the interest of the scientific community in recent years due to their properties and wide applications in photoelectrochemical cells, heterogeneous photocatalysis, optical switching, and transistors [1,2]. However, not only are nanoparticles applicable in this field, but they are also in health. The infectious diseases caused by resistant microorganisms are leading into pathologies of difficult therapeutic treatment [3,4]. Multi-drug resistant (MDR) microorganisms rise up due to misuse and abuse of antibiotics that favors the appearance of new genes [5]. This is important to note because one of most significant issues in public health are the nosocomial infections (NI) that occur in hospital settings [6]. In this sense, there are several alternatives using NPs in theses health issues, but it is important to note that all properties depend mainly on size, shape, geometry, and surface reactivity of nanomaterials [7,8,9,10].

In recent years, the use of NPs in medical applications is increasing because several pathogenic microorganisms are less able to develop resistance to NPs compared to common antimicrobial agents such as antibiotics and biomimetic models [11,12,13,14,15,16,17].

NPs can interact electrostatically with the bacteria membrane causing its alteration. Moreover, they can promote the formation of highly oxidant free radicals such as HO• and O_2_^•−^ [18,19,20,21]. These radicals cause secondary damage to the membrane and cell wall, hamper protein function, and destroy DNA [22,23]. A mechanism associated with the generation of free radicals is the photocatalytic activation of metalllic oxide nanoparticles. For instance, ZnONPs exhibit strong bactericidal properties when they are exposed to UV radiation [24]. The photocatalytic process of NPs is effective for the inactivation of both Gram- (e.g., *E. coli*, *L. monocytogenes*, *S. Enteritidis*) and Gram+ (e.g., *B. subtilis*, *S. aureus*, *S. pyogenes*, *E. fecalis*) bacteria [25,26]. The strongest bactericidal activity of the photocatalytic process is attributed to the action of hydroxyl radicals, which are particularly unstable and react quickly with most biological molecules, leading to cell death, among other effects [27,28].

MnO_2_ NPs have been reported as strong disinfectant agents that synergistically act through direct contact with the bacterial membrane, and to generate reactive oxygen species. These species can be produced in the presence of the active oxygen in the lattice as defects or adsorbed on the surface, and so MnO_2_ becomes an obvious choice as an oxidant in catalysis [29]. There have been reported that “the formation energy of oxygen vacancies at grain boundaries/edges is lower than that of bulk MnO_2_, and the reduction of Mn^4+^ to Mn^3+^ is also preferable at the boundaries/edges, which may provide a key point of the increased concentration of oxygen vacancies and unsaturated Mn^4+/^Mn^3+,^ both in favour of the abundant active sites for the electro-catalytic property”, which is an important fact for biological applications in disinfection [30]. On the other hand, nanostructured manganese dioxide particles when used in a photoelectrode can increase the utilization rate of visible light irradiation, due to its narrow band gap, large surface area, and negatively charged surface. However, the photocatalytic activity of MnO_2_ is still restricted by the slow rate of charge transfer and high recombination probability of photogenerated electron-hole pairs. Many efforts have been made to improve the photocatalytic capability of MnO_2_, a large number of studies showed that MnO_2_ combined with a conductor or semiconductor can effectively improve the photocatalytic activity compared to pure nanostructures.

Manganese oxides-based materials can be prepared by several methodologies such as hydrothermal, chemical bath deposition, sol-gel, electrodeposition, solvothermal, and co-precipitation. However, some of these classical methodologies are not environmentally friendly and they are highly time consuming. Appendix A presented the advantages of this methodology of synthesis with relation to top-down and bottom-up methods [30]. Then, the synthesis of nanoparticles using fast and “green” technologies is highly desirable for an easy translation of nanostructures to real-life applications. In this sense, the laser ablation method is a good way to synthesize MnO_2_ NPs. Indeed, we should mention that some previous works have reported the obtention of MnO_2_ NPs by laser ablation, where these NPs show promising activity compared to the standard antibiotic rifampicin [31].

Towards real-world applications, the post-separation of used NPs is a serious limitation that should be considered. Therefore, a third generation of photocatalysts entails the use of suitable supports having photocatalytic activity, which leads to an improved efficiency of the process. A conductive substrate that enhances the electron mobility along the horizontal film surface is preferred. The increased carrier mobility suppresses the undesirable electron–hole recombination, which results in better overall performance. In this sense, the FTO polymer, owing to its conductive and semiconductor properties, arises as one of the most promissory support materials. In fact, TiO_2_ deposited on FTO showed improved photocatalytic activity to dye discoloration [32].

In this work, for the first time MnO_2_ NPs are deposited on FTO via dip-coating and are evaluated in terms of their electronic, electrochemical, and disinfecting properties. Initially, several MnO_2_ NPs were prepared using different ablation times (5 min and 10 min) and laser energy (25 mJ, 50 mJ, and 80 mJ), characterized using UV-vis, FTIR, diffuse reflectance, Scanning Electron Microscopy (SEM), and magnetic susceptibility measurements. Then, NPs with the best physicochemical results were evaluated against *E. coli* and *S. aureus* in colloidal suspensions through photocatalysis. Later, the NPs are deposited on FTO and electronic/electrochemical and disinfecting properties were investigated.

## 2. Materials and Methods

### 2.1. Reagents

Manganese oxide (MnO_2_) was provided by Merck- Colombia. Fluorine-doped tin dioxide electrodes (FTO), Litium perchlorate (LiClO_4_)_,_ acid perchloric (HClO_4_) and reference electrode (Ag|AgCl|KCl (saturated)), Eosin methylene blue agar, and Manitol Salt were provided by Sigma-Aldrich- Colombia. CIP was purchased from Sigma-Aldrich (St Louis, MO, USA), including benzoquinone and sodium azide.

### 2.2. Synthesis of Colloidal MnO_2_ Nanoparticles by Laser Ablation

MnO_2_ nanoparticles were synthesized by laser ablation. A design of the pressing template in steel was made to provide the proper shape for the target substrate in the laser ablation. In a mortar, 10 g of MnO_2_ were homogenized and deposited in the mold. The mold loaded with the metal oxide was subjected to 2 tons m^−2^ of pressure (using a TESSI hydraulic press) for 2 min to obtain a solid target. Subsequently, a sintering process was carried out in a muffle (Thermolyne FB141M). The material in the template was heated for 12 h at different temperature rates according to references [29,30]. Then, the NPs were produced utilizing an Nd:YAG laser (1064 nm, 9 ns, and 10 Hz), which impinged on the target metal oxide (MnO_2_). Additionally, a lens with a focal length of 5 cm was used, the target surface was located at the focus, and the laser frequency on the target was kept constant without varying the target-lens distance or the laser energy. The NPs were collected in methanol (considering the solvent effect in the estimation of the bactericidal properties).

### 2.3. Characterization of Colloidal MnO_2_

For the determination of size and morphological properties SEM (JEOL- USA) was used and Element distribution map (EDX) of MnO_2_ interpreted. Transmission Electron Microscope (TEM) Tecnai F20 Super Twin TMP; field emission source, resolution of 0.1 nm at 200 Kv, maximum magnification at TEM 1.0 MX, GATAN US 1000XP-P camera, Gatan, Inc., Pleasanton, CA. EDX Oxford Instruments XMAX detector. STEM Analysis—FISCHIONE Instruments Model M3000 FP5360/22 HAADF Detector 120/200 kV. The estimation of the optical properties was carried out using a UV-Vis spectrophotometer (NANOCOLOR UV/VIS II) Macherey- Nagel, Tauc model was used to determine the bandgap value [31,32] according to Equation (1).
(1)$$\left(\alpha h\nu \right)\;2=\;\alpha {\left(h\nu -E_{g}\right)}^{\frac{1}{r}}\;$$
where *r* can be equal to 1/2 for direct allowed transitions, 3/2 for directly forbidden transitions, 2 for indirectly allowed transitions, and 3 for indirect forbidden transitions. 

IR analyses were carried out in an FTIR Spectrophotometer (Agilent Cary 630) using Attenuated Total Reflection (ATR) FTIR spectrophotometry. Magnetic susceptibility measurements were performed using a SQUID magnetometer (MANICS DSM8), with cryostatic cooling between 4–298 K. The effective magnetic moment was determined using a Johnson Matthey balance model MSB model MK II 13094- 3002 at room temperature using the Gouy method. In this case, the apparent change in the mass of the sample is in response to the repulsion or attraction by a region of high magnetic field between the poles and this value is the relationship with the electron disappear number [33]. The RAMAN spectra were collected on polyethylene terephthalate eroded with acetone, using a spectrophotometer DXR Smart Raman of Thermo Scientific in the presence of N_2_. 

### 2.4. This Involved the Deposition of Colloidal Nanoparticles on FTO As a Transparent Conductor

The suspended nanoparticles were deposited on fluorine-doped tin dioxide electrode (FTO) (2.33 Ω cm^−1^ on a glass substrate XOP Glass, 25 mm × 25 mm) by the dip-coating method; for deposition, 20 immersions were made for 1 min in the colloidal suspension of the nanoparticles and 15 sec of drying at room temperature. After finishing the immersion-drying cycles, a sintering process at 500 °C for 1 h in an atmosphere of oxygen was carried out. As it is well known, this type of system that uses FTO has a high electrochemical performance [34] and optoelectronical versatility [35,36].

### 2.5. Characterization for MnO_2_ Deposited on FTO

Spectroelectrochemical analyses of the deposited NPs were performed. The cell consisted of a 1 × 1 × 5 cm square glass tube sealed at one end. The FTO sheet covered with the nanoparticles was the working electrode, a platinum wire was used as the counter electrode, and saturated Ag/AgCl electrode was the reference electrode. As the supporting electrolyte, LiClO_4_ was used (0.1 M, acidified with HClO_4_ until pH 2.8). Redox properties of NPs deposited on FTO were analyzed through cyclic voltammetry (Metrohm Autolab PGSTAT302N). Cyclovoltammetry sweeps of 50 mV s^−1^ (between −2.0 and 2.0 V) were performed.

The optical bandgap and the thickness of the NPs deposited on FTO were established by measuring the absorption spectra between 300 and 700 nm. Subsequently, the electrochemical cell was placed in the optical path and the wavelength of maximum absorbance was determined by placing the potential in the dark state by chronoamperometric jump. Then, electrochromic analysis of the films was carried out with cycles of charge and discharge between the dark and light potential, and the percentages of transmittance at the maximum absorbance length were simultaneously established. The darkening time (tc) and whitening time (tb), the contrast (ΔT), the optical density (OD), and the efficiency of conversion of charge into transmittance (η) were determined.

### 2.6. Characterization of the Photocatalytic Properties of MnO_2_ Nanoparticles

To evaluate the antibacterial activity of NPs in colloidal suspension and deposited on FTO, photocatalytic processes were carried out and *E. Coli* ATCC 11,229 and *S. aureus* (ATCC 25923) were used as bacteria. The target microorganism was inoculated on an agar plate count, and the exponential growth (10^5^ CFU mL^−1^) was guaranteed by stabilizing the optical density at 0.5 in 540 nm, where CFU: colony-forming unit. In the case of the colloidal NPs, 5 mL of suspension was mixed with 0.1 mL of bacterial solution (10% NaCl, ~10^3^ CFU mL^−1^). Meanwhile, for the photocatalytic process with deposited NPs on FTO, this sheet was immersed in 5 mL of bacterial solution (10% NaCl, ~10^3^ CFU mL^−1^). The bacterial medium in the inoculum presented a value of pH 6.7, which did not change in the presence of the suspended nanoparticles.

The sample to be treated was irradiated within an aluminum reactor equipped with five UVA lamps (F8T/BLB, 60 W each one and main emission band at 368 nm) and constant stirring assisted with a vortex. The aluminum prevents the absorption of light and allows the enrichment of the reflections to the sample. The FTO sheet was exposed as perpendicular to the lamps system to ensure better interaction with the light. After 20 or 30 min of treatment, an aliquot (100 μL) of the treated sample was taken, serial dilutions were made, followed by seeding in agar on Petri dishes. The bacteria counting was done after 24 h of incubation at 37 °C, as shown in Appendix A. To evaluate the reactive oxygen species generated by the catalyst under this radiation, ciprofloxacin was used as the target molecule. The CIP elimination was realized in presence of three scavengers (benzoquinone, sodium azida, and isopropanol). The CIP was irradiated to same conditions in the disinfection process, with the final time as T = 30 min. Finally, the R = rate in scavenger presence/rate in scavenger absence was determined. The concentration of the pharmaceutical contaminants was monitored using an Agilent 1200 chromatograph, equipped with a LiChrospher^®^ RP-18 column (5 µm) and a UV detector (set at 279 nm for CIP, respectively). The injection volume was 20 µL and the eluents were acetonitrile and formic acid (10 mM). The flow rate of the mobile phase was 0.5 mL min^−1^.

## 3. Results and Discussion

### 3.1. Characterization of Synthesized MnO_2_ Nanoparticles in Colloidal Suspension

Figure 1 shows the UV-Vis absorption spectra for MnO_2_ NPs synthesized at three different laser energies (25, 50, and 80 mJ) and operation times (5 and 10 min). As can be seen, as the laser energy increased from 25 to 80 mJ, the absorption was more intense. Additionally, greater color (yellow-brown) intensity was observed to the naked eye when higher laser energy was used (Appendix A). The absorbance of colloidal NPs depends on the increase in the number of nanoparticles, and this is related to the increase in the ablation energy [37,38,39]. It can be noted that the higher absorption of manganese oxide appears in the range of 210–280 nm, which can be due to transition between bands. Souri et al. reported that the absorption around 280 nm is due to n → π* or π → π* transitions [40]. According to our results, for the MnO_2_ NPs synthesized at 25 mJ and 5 min, an absorption band at 210 nm was evident with an emerging band at less energy. Meanwhile, for the NPs prepared at 50 and 80 mJ, the absorption increases significantly and tends to be gaussian at longer wavelengths suggesting not only presence of Mn (IV) species in the suspension, but also the formation of MnO_2_ NPs, particularly when 80 mJ and 10 min are used [41].

In addition to the effect of the laser energy on the NPs formation, the influence of the ablation time was also studied (Appendix A). It is important to note that the absorbance of the nanoparticles was higher and increased as the ablation time increased from 5 to 10 min, with λ_max_ = 210 and 227, respectively, suggesting that there is a rise in the number of nanoparticles and they are a little bit larger [42]. On the other hand, the bandgap value of metal oxide NPs in colloidal solution was determined using the Tauc method, and the results are reported in Table 1. For all MnO_2_ NPs formed, bandgaps between 3.84 and 5.82 eV were obtained. For NPs synthesized at 25 mJ, the highest bandgap values were obtained (5.75–5.82 eV), and those NPs synthesized at 80 mJ had the lowest values (3.84–4.00 eV). The differences in the bandgap values suggest that the change in the laser energy and ablation time produce nanoparticles of different sizes. The bandgap values of NPs obtained in our systems suggest that a semiconductor behavior is reported for materials used in photovoltaic and photocatalysts. Therefore, these NPs could be potential materials for applications in solar cell and photocatalytic processes [43].

According to UV-vis and Tauc results, 50 and 80 mJ are the best conditions to form MnO_2_ NPs, but even though 80 mJ generates more NPs, they are less stable and collapse with time. FTIR spectra of MnO_2_ NPs obtained at 50 mJ and 10 min of laser ablation were measured (Figure 2). It can be observed that the bands between 470 and 798 cm^−1^, which correspond to α-MnO_2_ NPs, fit well to the characteristic vibrations of MnO_2_ reported in the literature [44]. In fact, several authors report that the band related to the Mn-O bond occurs around 500 cm^−1^ [45]. In this case, the vibration is corresponding to 485 cm^−1^. The broad band in the wavenumber range of 3000 to 3700 cm^−1^ could be assigned to the H-O bond of methanol. The bands between 2700 and 2900 cm^−1^ can be related to the C-H and the signals at ~1450 and 1050 cm^−1^ could design the C-O bond of adsorbed methanol. The methanol is the solvent used in the synthesis of nanoparticles, which is why vibrations are expected. The RAMAN spectra in the next figure showed three bands for MnO_2_ around 690 and 890 cm^−1^, varying only in width. The spectrum obtained from the synthesized MnO_2_ nanoparticles suggest that it corresponds to the α-MnO_2_ phase. To obtain the RAMAN spectrum, the material was deposited on polyethylene terephthalate and eroded with acetone.

Once the spectroscopic properties were determined, morphological characterization was performed using SEM. Figure 3A presents the SEM micrographs and the particle size distribution of MnO_2_ NPs formed at 50 mJ and 10 min of laser ablation. In the SEM micrographs, spherical MnO_2_ NPs were observed with a size distribution between 90–160 nm. The element distribution map (EDX) corresponding to A is MnO_2_ (O, 52.77%; Mn, 47.23%) (Appendix A). The percentages are within the range reported by Selvam et al. for this crystal structure. Figure 3B, image TEM is showed, in which nanoparticles are observed of ca. 25 nm width. Additionally, visualized high crystalline for α-MnO_2_ particles are shown, with planes associated with this structure. This result confirms the single-phase crystallinity of the α-MnO_2_ obtained by ablation laser synthesis [45].

In addition to morphological characterization, the magnetic properties of NPs were measured (Figure 4) using magnetic susceptibility at different temperatures and effective magnetic moments of the NPs removing the solvent. Figure 4A shows a maximum value between 150 and 170 K, suggesting that a transition from the paramagnetic to antiferromagnetic phase occurred [44,45]. Additionally, a magnetization value of 4.2 MB was calculated, corresponding to Mn (IV) in octahedral configuration. This information is consistent with the configuration and geometry for MnO_2_ NPs. The magnetization was also confirmed through the hysteresis loop magnetic measurement at room temperature (Figure 4B), achieving a value around 4 MB. Indeed, it is reported that the magnetization values of the MnO_2_ depend on the polymorphic form, and a magnetization value of 4.16 MB corresponds to α-MnO_2_ [46].

From the above results, MnO_2_ NPs, which formed at 80 mJ and 10 min, owing to its smaller size, lower bandgap, and good stability are interesting materials for photocatalytic applications.

### 3.2. Characterization of MnO_2_ NPs Deposited on FTO

The NPs synthesized at 50 mJ and 10 min of laser ablation, which showed a good potential for photocatalytic applications, were deposited on FTO by a dip-coating method. To estimate the thickness film deposited, transmittance spectra were measured (Figure 5). As can be seen, the curve of the maximum and minimum of the refractive indexes are shown, which is necessary to determine the film thickness by applying the Swanepoel method (Equation (2)), where λ_1_ corresponds to the wavelength with the lowest energy and λ_2_ the wavelength with the highest energy, for a consecutive peak and valley in the absorbance spectrum; n_1_ and n_2_ are the refractive indices, respectively [47]. As results, MnO_2_ NPs deposited on FTO substrates showed a high transmittance ranging from 92.5 to 96%, giving an average of 639 ± 45 nm of film thickness, equivalent to approximately six monolayers of MnO_2_ NPs with an average size distribution of 100 nm (Figure 5).
(2)$$d=\frac{\lambda_{1}\lambda_{2}}{2\left(\lambda_{1}n_{2}-\lambda_{2}n_{1}\right)}$$

For the NPs deposited on FTO, the bandgap was determined from the plot of (αην)^2^ vs. hν (Figure 6). The bandgap value was 2.82 eV for the deposited MnO_2_. It can be noted that the bandgap values for the deposited NPs were lower than those for the corresponding colloidal suspensions (Table 1), which means the deposited material was more susceptible to activation, for photovoltaic/photocatalytic processes, using UVA light, visible light, or even solar light. Our results are consistent with another study, which reports that films deposited under oxygen-rich conditions have narrow bandgaps and high electrical conductivities [48]. In fact, the conductive properties of the FTO polymer provide the possibility to absorb UV light and generate a heterojunction with semiconductors of metallic oxide, thus reducing the bandgap of the material.

On the other hand, to elucidate the redox properties of the MnO_2_ NPs deposited on FTO, cyclic voltammetries were carried out (Figure 7). The cyclic voltammogram of MnO_2_ did not show intermediate peaks of reduction in MnO_2_ to Mn (Figure 7), which started at −1.6 V onwards (Equation (3)), using Ag/AgCl as the reference electrode. This behavior is in agreement with the Pourbaix diagram of Manganese, where acidic media favors the reduction process. The oxidation of Mn to Mn^2 +^ (Equation (4)) was evidenced at −0.98 V and the oxidation of Mn^3+^ to Mn^4 +^ at 1.7 V (Equation (5)). These results were consistent with some the redox potentials reported for manganese [49]. The blank (i.e., the supporting electrolyte in the absence of MnO_2_) showed the characteristic signal of water oxidation at 1.66 V, whereas the water reduction overlaps with the reduction in MnO_2_. Additionally, the films have coloration at lower potentials than −1.5 V and higher than 1.5 V (as additional suggestion of the redox processes), and non-colored films were observable between −0.5 to 0.5 V. Additionally, we should mention that after −1.6 V, the reduction reaction of H^+^ to H_2(g)_ can occur (Equation (6)), which was confirmed by the formation of bubbles near the working electrode. Moreover, in the voltammogram for MnO_2_ deposited on FTO, the region between −0.9 and 0.9 V had no redox peaks, which is an indicator of pseudo-capacitance of the MnO_2_ NPs [50,51], and then the material could be potentially used in electrochemical supercapacitors [52].
MnO_2_ + 4H^+^ + 4e^−^ → Mn + 2H_2_O    E: −2.0V(3)
Mn → Mn^2+^ + 2e−           E: −0.98 V(4)
Mn^3+^ +2H_2_O → MnO_2_ + 2e−      E: 1.7 V(5)
2H^+^ +2e−→ H_2_             E: 0.0 V(6)

To complete the characterization of the NPs of MnO_2_ deposited in FTO, a spectroelectrochemical study was performed. Figure 8 shows the absorption spectrum, in open and closed circuits. The wavelength of maximum absorbance for the MnO_2_ coatings was 420 ± 15 nm. Compared to the colloidal suspension, the deposited MnO_2_ is absorbed in a wider range of wavelengths, thus increasing the absorption toward the visible light region. Furthermore, it can be mentioned that the spectrum shape and the absorbance maximum value were close to those reported in another for MnO_2_ deposited on an indium tin oxide electrode (ITO), suggesting the versatility of MnO_2_ that shows similar properties when deposited on diverse electrodes.

Applying a potential difference to the electrochemical cell between 0 and ±1.5 V generates a color change from very pale yellow to brown in the NPs of MnO_2_. This suggests a transition from Mn (III) to Mn (IV) and vice versa, and the pseudocapacitive behavior of the electrode made of MnO_2_ [45]. Moreover, the electrochromic analyses for the NPs of MnO_2_ deposited on FTO were carried out through the potential applied by charging and discharging in the light and dark regions (Figure 9). Thus, the coloring time (tc), bleaching time (tb), optical density (ΔOD), and efficiency (η) were determined, and they are reported in Table 2. The ΔOD was used to determine η. The deposited MnO_2_ nanoparticles presented a coloration efficiency of 36.66 ± 0.40 cm^2^ C^−1^. Such a coloration efficiency may arise from the layered structures of MnO_2_ NPs deposited on the FTO and the oxidation and reduction transitions of the Mn (III)/Mn (IV) couple (Appendix A). The electrode had moderate values of coloring and bleaching times (on the order of seconds) that led to long switching responses [53,54]. Additionally, the prepared material presented low optical contrast (12.35%). It can be remarked that the η value of our electrode was similar to those reported by other researchers for MnO_2_ deposited on ITO [55], and despite the long switching times and low optical contrast, our material could be utilized in electrochromic devices for specific applications [56].

### 3.3. Photocatalytic Study on E. Coli and S. aureus

After the determination of different physicochemical properties of NPs, the inactivation of two representative microorganisms (*E. coli* and *S. aureus*) in aqueous media was assessed. The intrinsic bactericidal activity of NPs (in suspension or deposited on FTO) and their action as photocatalysts (using UVA light) were considered. Moreover, the corresponding control subsystems (i.e., bacteria in dark with MnO_2_, and bacteria under UVA in absences of NPs) were tested (Appendix A).

Figure 10A shows the surviving population of microorganisms (in a Log-scale) after processes when the NPs in colloidal suspension were used. From Figure 10A, it can be remarked that the population of both target bacteria survived in the dark without light and without MnO_2_ (BK), suggesting that the osmotic pressure of the aqueous matrix had a low effect on tested microorganisms. The UVA light alone induced a small inactivating effect, which is associated with damages to bacteria by action on its DNA and enzymes or by triggering of internal oxidative stress [57]. In turn, the NPs of MnO_2_ in dark showed that ~1.8 Log-unit of the bacteria population survived after 20 min of interaction with the material, suggesting the intrinsic bactericidal activity of the MnO_2_ NPs in suspension. The bactericidal activity of the MnO_2_ NPs in dark could be attributed to external physical damage (e.g., destruction of lipid molecules and cell-wall of bacteria) and internal chemical damages (e.g., oxidative stress by reactive oxygen species produced inside bacteria) [58]. Remarkably, for the photocatalytic system, the lowest surviving population (~0.5 Log-unit) was obtained. In this case, the disinfection process involves the synergistic effect of light, NPs, and ROS produced from the photoactivation of MnO_2_ NPs. As mentioned earlier, MnO_2_ is a semiconductor, and under UVA irradiation the electrons from the valence band are excited to the conduction band, leading electron-hole pairs. The holes are able to oxidize H_2_O and OH^−^, forming HO^•^. In turn, the electrons can reduce dissolved O_2_, producing O_2_^•-^. The O_2_^•-^ may be protonated by H^+^ in water (depending on the pH of the system), forming hydroperoxyl radical (HO_2_^•^) that can be subsequently converted to H_2_O_2_. The formed H_2_O_2_ can be dissociated, by electrons in the conduction band, into more reactive hydroxyl radicals [58]. Radical species generated in the photocatalytic process can successively attack the membrane and cell wall, inactivating the bacteria which can even lead to leakages of cell content and bacterial death.

In the case of the inactivation of the target microorganisms by the MnO_2_ NPs deposited on FTO (Figure 10B), its intrinsic bactericidal activity in dark was lower than the obtained for the colloidal suspension. This decrease could be rationalized considering that when the MnO_2_ is deposited, the internal damages caused by MnO_2_ NP_S_ entry to the bacteria are limited, and only the external route is acting. Interestingly, the photocatalytic system using the NPs deposited on FTO exhibited higher disinfecting activity (i.e., a lower surviving population was found) than the colloidal suspension, which can be attributed to the lower bandgap material on FTO and then, a better use of the UVA light. In addition, in the system based on the colloidal suspension, a light scattering by the NPs is also possible; whereas, in the system with the MnO_2_ NPs deposited on FTO, the light scattering phenomenon is very low, making the photocatalytic process more efficient for inactivating the target bacteria. On the other hand, from Figure 10, it can also be noted that in both systems, the inactivating action on *E. coli* is higher than on *S. aureus* (one order of magnitude). For disinfection of 30 min, the total disinfection was obtained. The difference observed between the two considered microorganisms was attributed to their morphological features. As *S. aureus* is a Gram+ bacterium, it has a thick peptidoglycan layer, and does not have an outer lipid membrane or porins; whereas, *E. coli* (Gram-) has a thin peptidoglycan layer, an outer lipid membrane, and porins [59]. These results show that the damage is mainly given on the membrane. It is important to note that at least 70% of ROS targets are membrane proteins [59], and that gram negative bacteria have just a lipid membrane as a first barrier, and gram+ have peptidoglycan as a first barrier, mainly made up of carbohydrates forming a hetero-polymer. Thus, on one side, ROS cause oxidation of lateral chains of several amino acids such as proline, lysine, threonine, arginine, histidine, methionine, and cysteine. In fact, methionine and cysteine are especially susceptible to oxidative action of almost all ROS. On the other side, the membrane lipids are also oxidized by free radicals, giving a lipid peroxidation which causes stiffness in the membrane. This peroxidation of side chains generates a lipid radical (LOO.), this leads to endo-peroxides, and finally to the formation of malondialdehyde (MDA) and 4-Hydroxynonenal (4-HNE), that cause damage to DNA and vital proteins [60].

On the other hand, as can be seen in Figure 10C, a new set of experiments using the MnO_2_ NPs deposited on FTO were carried out at longer treatment times (30 min) for further investigations related to photocatalityc disinfection properties of materials. Interestingly, total disinfection of *E. coli* was reached at 30 min. In turn, more than 99% of *S. Aereus* was removed at the same time. In this sense, our results open a new way of opportunities to develop new biologically active materials applicable at several industrial and clinical environments (Appendix A). Finally, to determine the participation of reactive species in the photocatalytic disinfection, the ROS progress with specific scavengers was evaluated. In this study, a molecule sensitive (ciprofloxacine) to attack by hydroxyl radicals, super-oxo anion, and singlet oxygen (oxidation-associated species) was chosen. It was observed that the evolution of the degradation of the molecule against the scavengers depends on the radical generated. For benzoquinone, the degradation progress is hampered, suggesting a high presence of super-oxo radical anion. On the contrary, with isopropanol, sensitive to hydroxyl radical and sodium azide sensitive to singlet oxygen, there were no significant changes in the degradation progress. These results suggest that the action pathway of photochemically activated MnO_2_ occurs via super-oxo anion radical. The ratio (Rr) between the degradation rate in the presence and absence of each scavenger was determined, as shown in Figure 10D. A value of Rr equal to 1 means that the species has no participation in the degradation, whereas a value of Rr lower than 1 indicates that the species contributes to the CIP elimination [60].

## 4. Conclusions

In the present work, MnO_2_ NPs were effectively synthesized. The laser energy and ablation times influenced on the spectroscopic properties of colloidal NPs. The best condition framing high concentration, stability, and semiconductor behavior was 50 mJ and 10 min with a laser wavelength of 1064 nm. It was also found that the dip-counting method (a simple and low-cost technique) was efficient for the fabrication of thin films of MnO_2_ NPs on FTO, and such electrodes showed pseudocapacitive and electrochromic properties. Additionally, both colloidal suspension and deposited NPs presented intrinsic antibacterial activity against *E. coli* and *S. aureus.* Under UVA irradiation, colloidal and deposited NPs participated in a photocatalytic process (involving the action of photogenerated radicals) for the bacteria inactivation. However, in the photocatalytic process, MnO_2_ NPs deposited on FTO were slightly more efficient to inactivate the target bacteria, probably due to a limited electron-hole recombination and an incremented light absorption. The nature of the microorganism influenced the disinfecting action. Because of morphological characteristics, the Gram+ bacterium (*S. aureus*) had stronger protection than the Gram- bacterium (*E. coli)* against the attack of NPs or radical species generated during the photocatalytic process. The study of the mechanism of the reactive oxygen species generated by this manganese dioxide, allows us to suggest that it is the super-oxo radical anion that is responsible for the microbial attack. More in situ studies on membrane permeation are needed to adequately understand and elucidate the mechanism of action.

## Figures and Tables

**Figure 1 nanomaterials-12-04061-f001:**
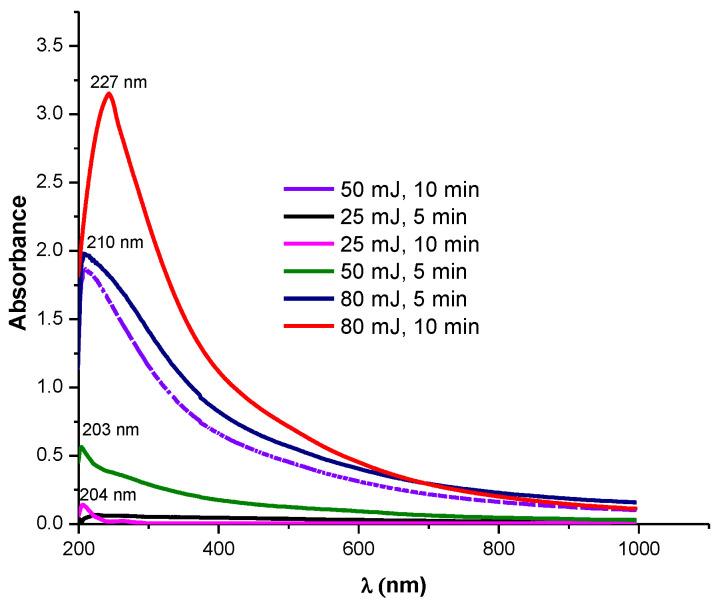
UV-vis spectra of NPs in colloidal suspension obtained at diverse laser ablation energies and 5 min and 10 min of ablation.

**Figure 2 nanomaterials-12-04061-f002:**
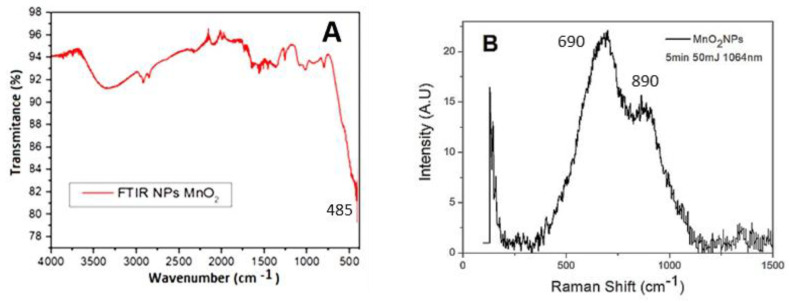
(**A**) FTIR, and (**B**) RAMAN spectrum for NPs in colloidal suspension.

**Figure 3 nanomaterials-12-04061-f003:**
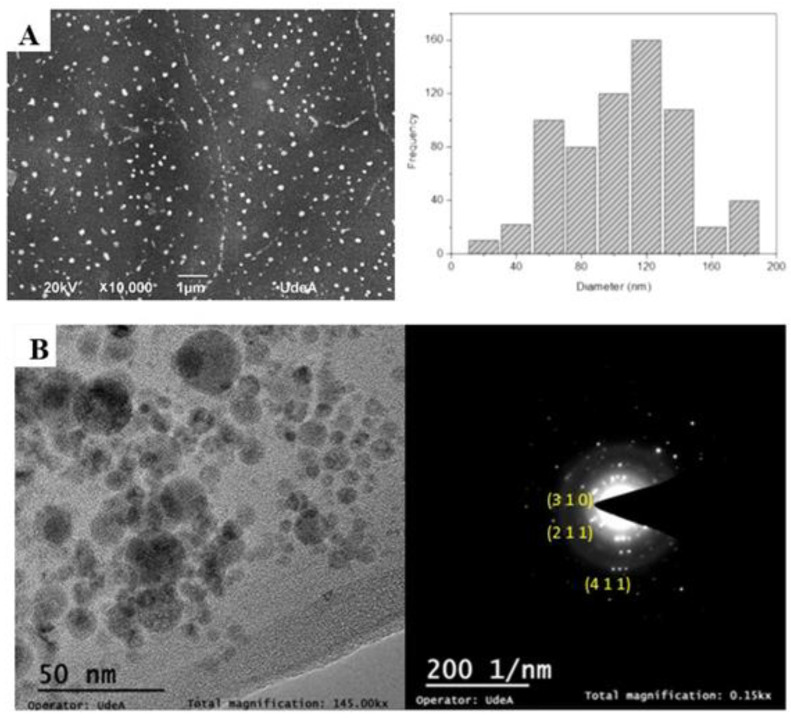
(**A**) SEM micrograph and particle size distribution of MnO_2_ NPs in colloidal solution formed at 50 mJ and 10 min; (**B**) TEM image for 50 nm and 200 1/nm for and XRD diagram.

**Figure 4 nanomaterials-12-04061-f004:**
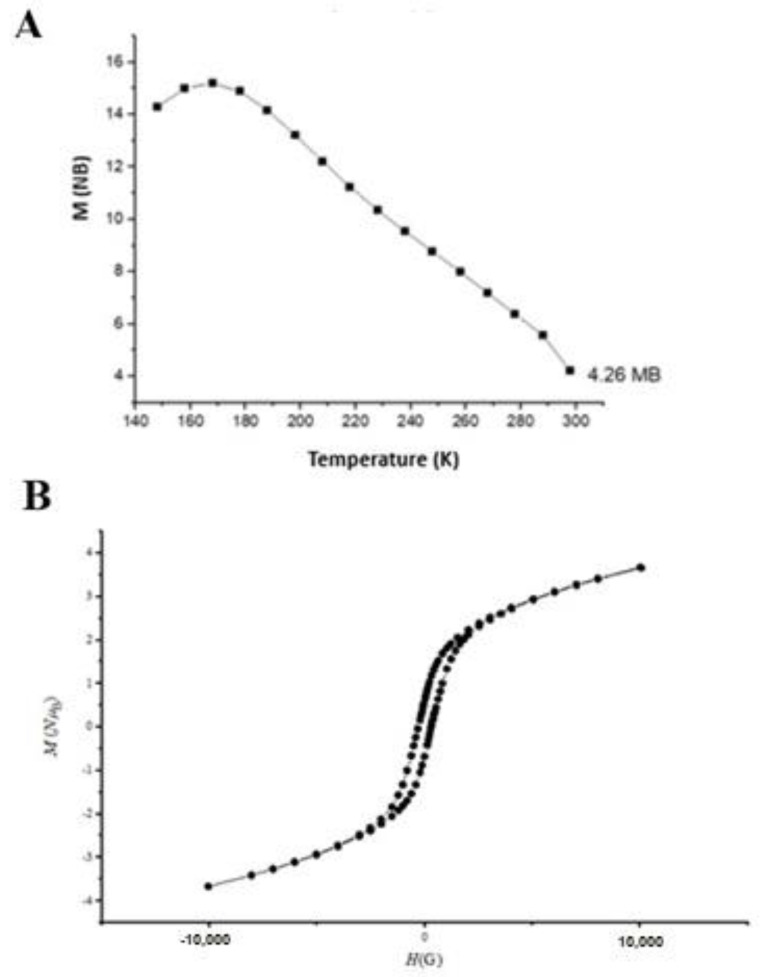
Magnetic properties of colloidal MnO_2_ NPs. (**A**) Susceptibility at different temperatures; (**B**) Magnetic hysteresis.

**Figure 5 nanomaterials-12-04061-f005:**
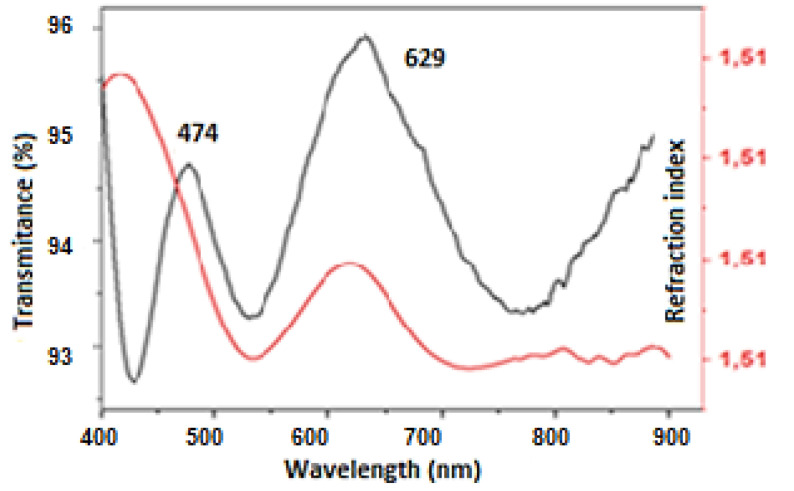
Transmittance spectra for NPs deposited on FTO.

**Figure 6 nanomaterials-12-04061-f006:**
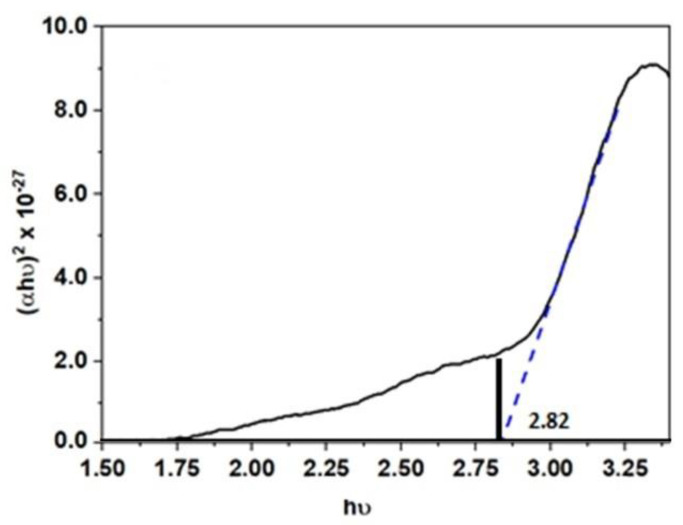
Bandgap determination of the MnO_2_ NPs deposited on FTO, blue line (representation of the graphical intersection on the x-axis (hν), as an estimate of the band gap value).

**Figure 7 nanomaterials-12-04061-f007:**
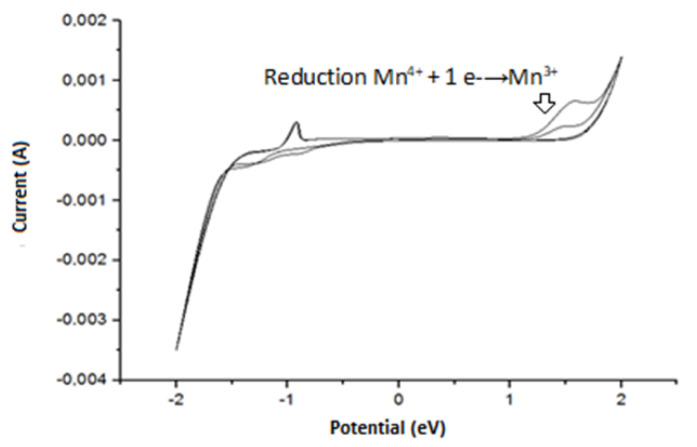
Cyclic voltammetry of NPs deposited on FTO. Scanning speed: 50 mV s^−1^, 0.1 M LiClO_4_ electrolyte, acidified with HClO_4_ at pH 2.80.

**Figure 8 nanomaterials-12-04061-f008:**
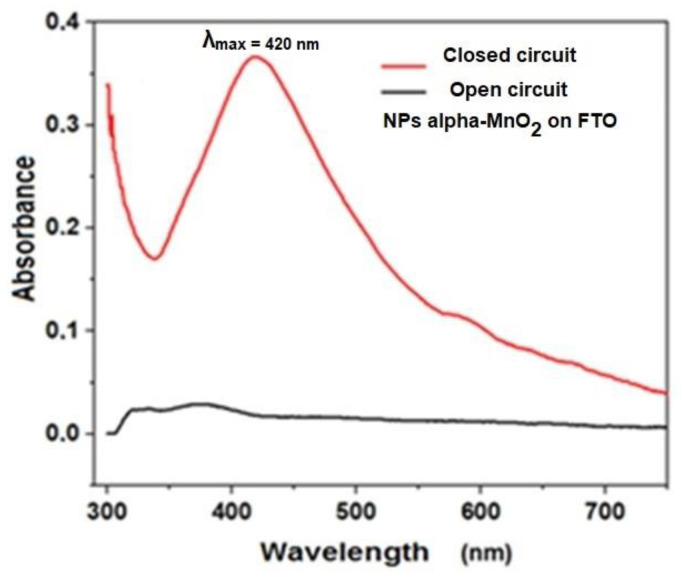
Absorption spectrum of NPs films deposited on FTO.

**Figure 9 nanomaterials-12-04061-f009:**
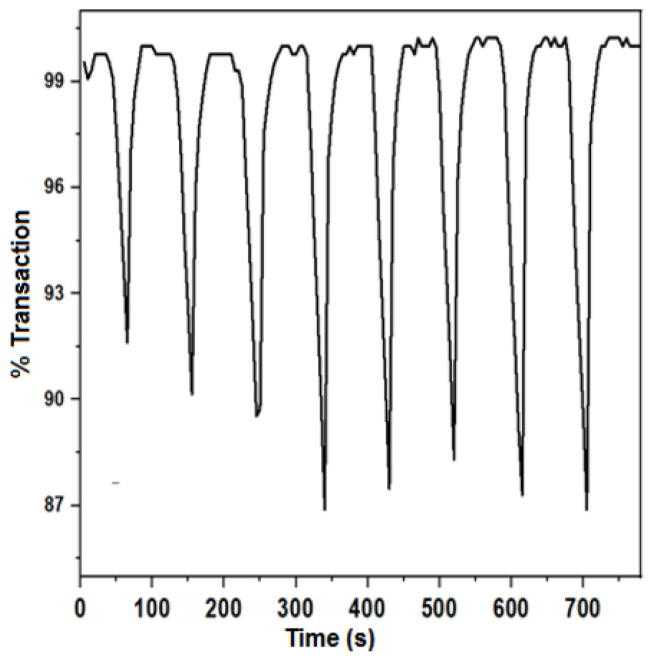
Electrochromic response of deposited NPs. Charge and discharge between 0 and 1.5 V vs. Ag/AgCl, LiClO_4_ electrolyte (0.1 M), acidified with HClO_4_ at pH 2.80.

**Figure 10 nanomaterials-12-04061-f010:**
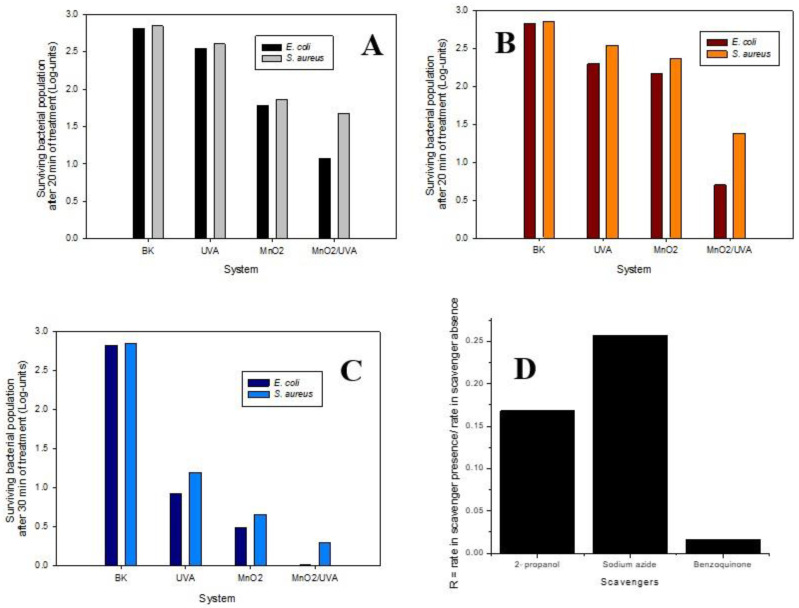
Photocatalytic process for bacteria inactivation using NPs. (**A**) Disinfection with NPs in colloidal suspension; (**B**) Disinfection with NPs deposited on FTO for t = 20 min; (**C**) Disinfection with NPs deposited on FTO for 30 min of photocatalytic treatment; (**D**) Scavengers R = rate in scavenger presence/rate in scavenger absence, effect of scavengers on the degradation of LEV. Experimental conditions: [CIP] = 30.6 µM, [IPA] = [SA] = [BQ] = 1.38 mM, [MnO_2_] = 0.01 g. L^−1^, exposition time = 30 min (UV radiation).

**Table 1 nanomaterials-12-04061-t001:** Bandgap values of MnO_2_ NPs in colloidal suspension.

Laser Energy (mJ)	Ablation Time (min)	Bandgap (eV)
25	5	5.75
	10	5.82
50	5	5.61
	10	4.23
80	5	4.00
	10	3.84

**Table 2 nanomaterials-12-04061-t002:** Electrochromic results of MnO_2_ NPs deposited in FTO.

Coloring Time (tc, in s)	Bleaching Time (tb, in s)	Contrast (ΔT, in%)	Optical Density (ΔOD)	Efficiency (η, in cm^2^ C^−1^)
28.2 ± 1.9	22.9 ± 3.1	12.35 ± 0.61	0.059 ± 0.003	36.66 ± 0.40

Average charge: 4.06 ± 0.4 mC, average area: 2.5 ± 0.03 cm^2^.

## Data Availability

Not applicable.

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
