# Peer review of "Manganese Dioxide Nanoparticles Prepared by Laser Ablation as Materials with Interesting Electronic, Electrochemical, and Disinfecting Properties in Both Colloidal Suspensions and Deposited on Fluorine-Doped Tin Oxide"

_nanomaterials, 2022, doi:10.3390/nano12224061_

Round 1
Reviewer 1 Report
Comments to the authors:
This manuscript reported the synthesis of MnO2 nanoparticles by laser ablation and dip-coating deposition on FTO. And the optical, spectroscopic and electrochemical proprieties were carefully investigated, the MnO2 NPS on FTO by deposition method exhibited the smaller band gap than that by colloidal method. Importantly, the MnO2 NPS on FTO possessed an excellent antibacterial activity against E. coli and S. aureus, especially in the presence of UVA light. Thus, both the colloidal suspension and deposited NPs can act as disinfecting agents themselves or be activated by light. The manuscript is well-organized and the data presented are of high quality. The topic of the work is significant with regard to the development of photocatalysis technology. The manuscript can be considered for publication after the following comments are addressed. 1. The authors discussed the mechanism of antibacterial activity with light irradiation, as far as we know, some radicals can be detected by EPR during the reaction, so the authors should make a deep discussion for the mechanism with the help of EPR, and the quality of this work will be greatly improved. |
Author Response
We re-submit hereby the manuscript entitled: Manganese dioxide nanoparticles prepared by laser ablation as materials with interesting electronic, electrochemical, and disinfecting properties in colloidal suspension and deposited on FTO.
Thank you very much to the Editor and reviewers for the valuable corrections/suggestions made to our manuscript, which have been very useful to improve the quality of the document. Below you can find the authors answer to each one of the addressed issues.
Reviewer 1:
This manuscript reported the synthesis of MnO2 nanoparticles by laser ablation and dip-coating deposition on FTO. And the optical, spectroscopic and electrochemical proprieties were carefully investigated, the MnO2 NPS on FTO by deposition method exhibited the smaller band gap than that by colloidal method. Importantly, the MnO2 NPS on FTO possessed an excellent antibacterial activity against E. coli and S. aureus, especially in the presence of UVA light. Thus, both the colloidal suspension and deposited NPs can act as disinfecting agents themselves or be activated by light. The manuscript is well-organized and the data presented are of high quality. The topic of the work is significant with regard to the development of photocatalysis technology. The manuscript can be considered for publication after the following comments are addressed.
- The authors discussed the mechanism of antibacterial activity with light irradiation, as far as we know, some radicals can be detected by EPR during the reaction, so the authors should make a deep discussion for the mechanism with the help of EPR, and the quality of this work will be greatly improved.
Answer: In this work is suggest the formation of ROS species. The holes are able to oxidize H2O and OH−, forming HO•. In turn, the electrons can reduce dissolved O2, producing O2•-. The O2•- may be protonated by H+ in water (depending on the pH of the system), forming hydroperoxyl radical (HO2•) that can be subsequently converted to H2O2. The formed H2O2 can be dissociated, by electrons in the conduction band, into more reactive hydroxyl radicals. As is mentioned by the Reviewer EPR can be carried in order to detect ROS using specifically spin trap.
Unfortunately, we do not have nationwide access to EPR. We established collaboration with UNAM in Mexico, but sending samples, setting up spin trap conditions and acquiring spin traps have been difficult to achieve at this time. However, as we consider that it is very important to propose the ROS produced under photocatalysis by the NPs MnO2, we have done the following. A drug (cirpofloxacin) sensitive to hydroxyl radical, super-oxo anion and singlet oxygen was chosen. Subsequently, in the presence of Nps MnO2, selective scavengers were added for each ROS. It was irradiated at the same wavelength and the decay in the production of a specific ROS by the scavenger suggest its presence in the medium.
In the manuscript this information was add as follow:
2.6. Characterization of the photocatalytic properties of MnO2 nanoparticles
To evaluate the reactive oxygen species generated by the catalyst under this radiation, ciprofloxacin was used as the target molecule. The CIP elimination was realized in presence of three scavenger (benzoquinone, sodium aside and isopropanol). The CIP was irradiated to same conditions in the disinfection process, witn final time T= 30 min. Finally the R = rate in scavenger presence/ rate in scavenger absence was determinated. The concentration of the pharmaceutical contaminants was monitored using an Agilent 1200 chromatograph, equipped with a LiChrospher® RP-18 column (5 µm) and a UV detector (set at 279 nm for CIP, respectively). The injection volume was 20 µL and the eluents were acetonitrile and formic acid (10 mM). The flow rate of the mobile phase was 0.5 mL min -1 .
Finally, to determine the participation of reactive species in the photocatalytic desinfection, it was evaluated the ROS progress with specific scavengers. In this study, a molecule sensitive (ciprofloxacine) to attack by hydroxyl radicals, super-oxo anion and singlet oxygen (oxidation-associated species) was chosen. It was observed that the evolution of the degradation of the molecule against the scavengers depends on the radical generated. For benzoquinone the degradation progress is hampered, suggesting high presence of super-oxo radical anion. On the contrary, with isopropanol, sensitive to hydroxyl radical and sodium aside sensitive to singlet oxygen, there were no significant changes in the degradation progress. These results suggest that the action pathway of photochemically activated MnO2 occurs via super-oxo anion radical. Ratio (Rr) between the degradation rate in the presence and absence of each scavenger was determined, as shown the Figure 10 D. A value of Rr equal to 1 means that the species has no participation in the degradation, whereas a value of Rr lower than 1 indicates that the species contributes to the CIP elimination.

Reviewer 2 Report
Report on manuscript (Nanomaterials) – 1928601
"Manganese dioxide nanoparticles prepared by laser ablation as materials with interesting electronic, electrochemical, and disinfecting properties in colloidal suspension and deposited on FTO"
by Jhonatan Corrales, Jorge Acosta-Vergara, Sandra Castro, Henry Riascos, Efraim Serna- Galvis, Ricardo Torres-Palma and Yenny Ávila- Torres
intended for publication in Nanomaterials
Summary
This manuscript intended for publication in Nanomaterials as a regular article provides an overview on new results of the production of MnO2-nanoparticles using laser ablation with various wavelengths. The nanoparticles were deposited by dip-coating on fluorine-doped tin oxide. Several physico-chemical characterizations were performed: optical and spectroscopic properties have been determined as well as electro-chemical studies were carried out. Key applications were disinfecting properties studied in colloidal and deposited regimes using E. coli and S. aureus microorganisms. Although the experiments are well described, some passages need further detailed explanation (see the following remarks). Generally, the number of references is too large for an article. The style is in a way rather descriptive and should be more compact and quantitative. Maybe, the use of some more tables will improve the readability.
In summary, after major revision, I recommend this work for publication in Nanomaterials.
List of remarks and corrections
· title: I would recommend to write out the word “fluorine-doped tin oxide” instead of FTO in the title, because this acronym is only known in a certain community
· abstract, page 1, line 23: band gap (typo)
· introduction, page 1, line 43: what does [4] refer to? The references are numbered by small letters
· introduction, page 2, line 54: exhibit (typo)
· introduction, page 2, line 61: …NPs have… (typo)
· introduction, page 2, line 63: …prepared by… (please insert a blank)
· introduction, page 2, line 84: …laser energy… instead of …energy laser…
· 2.3 Characterization of colloidal MnO2, page 3, line 122: please briefly explain Gouy method in the text
· page 3, line 127: please clarify the unit sq-1 (this seems to be no SI unit)
· page 4, line 155, please explain “CFU”
· page 4, line 160: please indicate the reason for using an aluminum reactor
· page 4, line 175: what does [31] refer to? The references are numbered by small letters
· page 4, line 180: significantly (typo)
· page 5, fig. 1: I would recommend to combine both graphs in a single one to emphasize the changes of absorbance when altering the ablation duration
· page 5, line 192: what does [31], [40] refer to? The references are numbered by small letters
· page 5, line 193: please explain briefly the “Tauc method” in the text
· page 5, line 203: “Tauc” results (typo)
· page 5, line 208: …reported in the literature…Here, the respective reference(s) should be mentioned
· page 7, fig 4: the size of the letters (x- and y-axis) is too small; the graphs have a poor technical quality
· page 8, line 273: electrode (typo)
· page 9, line 298: what does [61] refer to? The references are numbered by small letters
· page 9, line 303: what does [60] refer to? The references are numbered by small letters
· page 11, line 344, please delete …the…(typo)
· page 11, line 367: …E. coli is higher than…
What do you exactly mean with higher? Please quantify.
· page 12, Fig. 10: the indication of A, B, C is missing
· page 12, line 398: please write:…laser wavelength of 1064 nm…
references - general remark: it is difficult to find the respective references using this system of lettering (e.g. xxxi); it would be more convenient to use numbers like [1], [2]…
Author Response
We re-submit hereby the manuscript entitled: Manganese dioxide nanoparticles prepared by laser ablation as materials with interesting electronic, electrochemical, and disinfecting properties in colloidal suspension and deposited on FTO.
Thank you very much to the Editor and reviewers for the valuable corrections/suggestions made to our manuscript, which have been very useful to improve the quality of the document. Below you can find the authors answer to each one of the addressed issues.
This manuscript intended for publication in Nanomaterials as a regular article provides an overview on new results of the production of MnO2-nanoparticles using laser ablation with various wavelengths. The nanoparticles were deposited by dip-coating on fluorine-doped tin oxide. Several physico-chemical characterizations were performed: optical and spectroscopic properties have been determined as well as electro-chemical studies were carried out. Key applications were disinfecting properties studied in colloidal and deposited regimes using E. coli and S. aureus microorganisms. Although the experiments are well described, some passages need further detailed explanation (see the following remarks). Generally, the number of references is too large for an article. The style is in a way rather descriptive and should be more compact and quantitative. Maybe, the use of some more tables will improve the readability.
In summary, after major revision, I recommend this work for publication in Nanomaterials.
List of remarks and corrections
- title: I would recommend to write out the word “fluorine-doped tin oxide” instead of FTO in the title, because this acronym is only known in a certain community
Answer: The title was changed as follow: Manganese dioxide nanoparticles prepared by laser ablationas materials with interesting electronic, electrochemical, and disinfecting properties in both colloidal suspensions and deposited on fluorine-doped tin oxide
- abstract, page 1, line 23: band gap (typo):
Answer: The line 23 was reformed as follow: For the deposition of NPs on FTO sheet is produced an anode with a pseudocapacitive behavior, 2.82 eV of bang gap (GAP) in comparison with colloidal NPs for a value of 3.84 eV.
- introduction, page 1, line 43: what does [4] refer to? The references are numbered by small letters
Answer: This reference had a numbering error, the [4] in the line 43 was retired.
- introduction, page 2, line 54: exhibit (typo), introduction, page 2, line 61: …NPs have… (typo), introduction, page 2, introduction, page 2, line 84: …laser energy… instead of …energy laser…
Answer: All corrections were incorporated to manuscript.
- 3 Characterization of colloidal MnO2, page 3, line 122: please briefly explain Gouy method in the text.
Answer: We briefly explain Gouy method as follow: In this case, the apparent change in the mass of the sample is response to repulsion or attraction by a region of high magnetic field between the poles and this value is relationship. with the electron disappear number
- page 4, line 155, please explain “CFU”
Answer: The CFU was explain as follow: The target microorganism was inoculated on agar plate count, and the exponential growth (105 CFU mL-1) was guaranteed by stabilizing the optical density at 0.5 in 540 nm, where CFU: colony-forming unit.
- page 4, line 160: please indicate the reason for using an aluminium reactor.
Answer: The aluminum prevents the absorption of light and allows the enrichment of the reflections to the sample.
- page 4, line 175: what does [31] refer to? The references are numbered by small letters.
Answer: This reference had a numbering error, the [31] in the line 175 was retired.
- page 5, line 192: what does [31], [40] refer to? The references are numbered by small letters.
Answer: This reference had a numbering error, the [31] and [40] was retired.
- page 5, line 193: please explain briefly the “Tauc method” in the text.
Answer: The Tauc method was explained in the section Characterization of colloidal MnO2, as follow:
Tauc model was used to determine the band-gap value according to Eq. 1.
(Eq. 1) |
Where r can be equal to 1/2 for direct allowed transitions, 3/2 for directly forbidden transitions, 2 for indirectly allowed transitions, and 3 for indirect forbidden transitions [¡Error! Marcador no definido.].
- What do you exactly mean with higher? Please quantify.
Answer: The higher was specificized as: is higher than on S. aureus (one Log unit). For disinfection of 30 min the total disinfection was obtained.
- page 12, line 398: please write:…laser wavelength of 1064 nm…
Answer: The text was re- written as suggest to evaluator.

Reviewer 3 Report
This manuscript uses the sample of manganese dioxide nanoparticles deposited on FTO prepared by laser ablation to evaluate the electronic, electrochemical and antibacterial activity. The material innovation is general, but the overall work is relatively solid, and the research work is worthy of publication on Nanomaterials. The manuscript has some problems, and it has the potential to be published after major revision.
1. The author needs to further link the material structure characterization with antibacterial activity, connecting the whole work and sublimation. Why does the author choose this material for antibacterial modeling, is it accidental discovery or based on previous work, which has reference significance for other work?
2. Why does the author use laser ablation to prepare MnO2 instead of hydrothermal or other simple methods? Although the author says that other methods are not environmentally friendly and time-consuming, please use tabular data to explain, and the laser ablation method is not a worthwhile preparation for equipment requirements or cost. Is the material synthesized by this method highly active?
3. Please give the pH of bacterial culture. Fig 2 Main peak suggested to be added for analysis.
4. Figure 3 is lack of EDX element distribution map of MnO2, and should be supplemented TEM diagram for its morphology observation is more obvious, prove the successful synthesis of MnO2, also need to add transmission electron microscopy diffraction and XRD diagram to prove that the synthetic material is single crystal or polycrystalline.
5. The author should draw a flow chart of the preparation of the material MnO2 to the sterilization mechanism, which is a clearer idea and helps to improve the readability of the article.
6. Line 300: Color changes require more direct evidence as supporting material (e.g. video form).
7. The sterilization process involves a synergistic effect of light, NPs, and ROS produced by the photoactivation of MnO2 NPs, but which one is the most important is illustrated with experimental data.
8. The illustration in Figure 10 has a problem and then no A, B, C.
9. The authors should expand the sample size of the data to make the antibacterial effect of MnO2 more realistic, without accidental errors.
10. The authors should compare the antibacterial properties of MnO2 with traditional antibiotics or other antibacterial organisms and carbon dots.
11. Reference format needs to be carefully checked and corrected.
Author Response
We re-submit hereby the manuscript entitled: Manganese dioxide nanoparticles prepared by laser ablation as materials with interesting electronic, electrochemical, and disinfecting properties in colloidal suspension and deposited on FTO.
Thank you very much to the Editor and reviewers for the valuable corrections/suggestions made to our manuscript, which have been very useful to improve the quality of the document. Below you can find the authors answer to each one of the addressed issues.
His manuscript uses the sample of manganese dioxide nanoparticles deposited on FTO prepared by laser ablation to evaluate the electronic, electrochemical and antibacterial activity. The material innovation is general, but the overall work is relatively solid, and the research work is worthy of publication on Nanomaterials. The manuscript has some problems, and it has the potential to be published after major revision.
- The author needs to further link the material structure characterization with antibacterial activity, connecting the whole work and sublimation. Why does the author choose this material for antibacterial modeling, is it accidental discovery or based on previous work, which has reference significance for other work?
Answer: These species can be produced in presence of the active oxygen in the lattice as defects or adsorbed on the surface, and so MnO2 becomes an obvious choice as an oxidant in catalysis. There have been reported that “the formation energy of oxygen vacancies at grain boundaries/edges is lower than that of bulk MnO2, and the reduction of Mn4+ to Mn3+ is also preferable at the boundaries/edges, which may provide a key point of the increased concentration of oxygen vacancies and unsaturated Mn4+/Mn3+, both in favor of the abundant active sites for the electro-catalytic property”, important fact for biological applications in disinfection. On the other hand, nanostructured manganese dioxide particles when used in a photoelectrode can increase the utilization rate of visible light irradiation, due to its narrow band gap, large surface area and negatively charged surface. But the photocatalytic activity of MnO2 is still restricted by the slow rate of charge transfer and high recombination probability of photogenerated electron-hole pairs. Many efforts have been made to improve the photocatalytic capability of MnO2, a large number of studies showed that MnO2 combined with a conductor or semiconductor can effectively improve the photocatalytic activity compared to pure nanostructures.
- Why does the author use laser ablation to prepare MnO2instead of hydrothermal or other simple methods? Although the author says that other methods are not environmentally friendly and time-consuming, please use tabular data to explain, and the laser ablation method is not a worthwhile preparation for equipment requirements or cost. Is the material synthesized by this method highly active?
Answer: The authors use laser ablation as a synthesis method for the following conditions:
Fewer reagents are used in the synthesis. The influence of temperature was considered in the sintering of the target. Once the target has been made, the synthesis is fast and simple, simply modulating the variables of fluence, energy and time. The best disinfectant properties were obtained in suspension of methanol, not water. The method used allowed to evaluate the effect of the polarization of the solvent. Although the cost of the laser is significant at the beginning of the technique, once this investment is made, the cost per analysis is small and allows manipulation in suspension or deposition under similar conditions. Finally, it was indeed the method that allowed the best stabilization with methanol.
Following the suggestions of the evaluators, a table is made with the different advantages in the synthesis methods, evidenced in terms of clean synthesis to laser ablation as one of the best, Table S1.
Table S1. Ventages in top down and bottom- up methods for synthesis of nanoparticles
Top down methods |
Bottom up methods |
|
||||
|
Ventages |
|
Ventages |
|
||
Mechanical milling |
Ball milling |
Enabling chemical reactions in the absence of solvents, therefore resulting in a cleaner synthetic procedure. Obtaining products that are difficult or impossible to obtain in the presence of bulk solvents, and expanding the scope of chemical reactions to slightly soluble or inert starting materials.
|
Solid state methods |
Physical vapor deposition |
Particle properties of nanostructures such as surface morphology and crystal structure can be controlled.
Chemical vapor deposition method of coating exhibits the high film durability.
This method is easy to scale-up.
Produces nanoparticles of controlled surface morphology. |
|
2 |
Laser ablation |
It is easy to obtain multi- component film that is of the desired stoichiometric ratio by PLD.
It has high deposition rate, short test period and low substrate temperature requirements.
Films prepared by laser ablation are uniform.
The process is simple and flexible with great development potential and great compatibility
Process parameters can be arbitrarily adjusted, and there is no limit to the type of targets.
Multi-target components are flexible, and it is easy to prepare multilayer films and heterojunctions.
It is easy to clean and can prepare a variety of thin film materials.
Use UV pulsed laser of high photon capability and high energy density as the energy source for plasma generation, so it is non-polluting and easy to control.
|
Liquid state synthesis methods: Sol gel methods |
Simple method for the formation of thin metal films.
Particle size and morphology is possible to control by systematic monitoring of reaction parameters
Desired size and shape nanoparticle can be prepared.
Well-crystallized powder can be formed.
Produce nanocrystal with high crystallinity. |
|
|
3 |
Sputtering |
The composition of sputtered material is not altered and remains same as that of the target material. Method of choice for refractory metals and intermetallic compounds than other methods like evaporation and laser ablation.
Economical method as the sputtering equipment is less expensive than electron-beam lithography systems.
Less impurities are generated than those created by chemical methods.
Alloy nanoparticles can be produced with easier control on composition than other chemical reduction methods.
This method is a versatile technique to synthesize ionic nanoparticles with spacious sizes and compositions that are not obtainable in solution.
Slow deposition of heavier ions or mass-selected ions gives unparalleled control of different parameters such as size, composition and charges of ions deposited onto surfaces. |
Gas phase methods Spray pyrolysis |
Relatively simple method.
Low cost method.
The particle size can be controlled and reproducible. |
|
- Please give the pH of bacterial culture. Fig 2 Main peak suggested to be added for analysis.
Answer: The pH of bacterial culture was 6.7. This value was incorporated al article in the section 2.6. Characterization of the photocatalytic properties of MnO2 nanoparticles, as follow:
“The bacterial medium in the inoculum presented a value of pH 6.7, which did not change in the presence of the suspended nanoparticles”
In the Fig 2. The main peak in the IR and RAMAN spectra were added, as follow:
Figure 2. A) FTIR and, B) RAMAN spectrum for NPs in colloidal suspension.
- Figure 3 is lack of EDX element distribution map of MnO2, and should be supplemented TEM diagram for its morphology observation is more obvious, prove the successful synthesis of MnO2, also need to add transmission electron microscopy diffraction and XRD diagram to prove that the synthetic material is single crystal or polycrystalline.
Answer: The EDX elemental distribution map of MnO2 was added in supplementary information. The element distribution map (EDX) corresponding to a- MnO2 (O, 52.77%; Mn, 47.23%). The percentage corresponding to MnO2 structure.
Figure S4. Element distribution map (EDX) of MnO2 for SEM micrograph.
The Image TEM showed nanoparticles of 20 nm width. Also visualized high crystalline for α-MnO2 particles, with planes associated to this structure. This result confirms the single-phase crystallinity of the α-MnO2 obtained by ablation laser synthesis.
Figure 3. (A) SEM micrograph and particle size distribution of MnO2 NPs in colloidal solution formed at 50 mJ and 10 min. (B) TEM image for 50 nm and 200 1/nm for and XRD diagram
- The author should draw a flow chart of the preparation of the material MnO2to the sterilization mechanism, which is a clearer idea and helps to improve the readability of the article.
Answer: The authors added the flow chart of the preparation of the material MnO2 sterilization mechanism in supplementary information, as follow:
Figure S5. flow chart of the preparation of the material MnO2 sterilization mechanism
- Line 300: Color changes require more direct evidence as supporting material (e.g.video form).
Answer: As evidence of the color changes, samples of the electrochemical process have been taken. Discoloration is visualized in the images, for all conditions. However, it was not possible to make the video since the cell is attached to the measuring instrument and is ensembled and covered.
Figure S6. Evidence of color change in suspended Nps from MnO2 (M1: 80mJ, 10min; M2: 80mJ, 5 min; M3: 50mJ, 10 min; M4: 50mJ, 5 min; M5; 25mJ, 10 min; M6: 25mJ, 5 min) after reduction to Mn3+ for electrochromic analyses.
- The sterilization process involves a synergistic effect of light, NPs, and ROS produced by the photoactivation of MnO2NPs, but which one is the most important is illustrated with experimental data.
Answer: Indeed, the graphs only show the final effect of photocatalysis. However, each test of the effect of the adsorption of the microorganism on the Nps and the effect of photolysis was considered. These contributions were subtracted from the photocatalysis data. In that sense, the tables with the graphed values are attached.
Table S1. Photocatalytic process for bacteria inactivation using NPs for t= 20 min (suspension, deposited) and 30 min (deposited) of photocatalytic treatment, Units Log UFC
Method |
Control BK |
Interaction UV |
Visible Photolysis |
Adsorption |
MnO2 (UV) |
MnO2 (VIS) |
|||
Exposition T=20 min |
Suspension Nps |
E.Coli |
2.82 |
2.54 |
2.48 |
1.78 |
1.08 |
1.69 |
|
S.Aureus |
2.85 |
2.60 |
2.47 |
1.85 |
1.67 |
1.39 |
|||
Deposited NPs |
E.Coli |
2.82 |
2.29 |
2.52 |
2.16 |
0.69 |
1.50 |
||
S.Aureus |
2.85 |
2.53 |
2.37 |
2.36 |
1.38 |
1.25 |
|||
Exposition T=30 min |
Deposited Nps |
Method |
Control BK |
Interaction UV |
Adsorption |
MnO2/UVA |
|||
E.Coli |
2.82 |
0.93 |
0.49 |
0 |
|||||
S.Aureus |
2.85 |
1.2 |
0.66 |
0.3 |
|||||
- The illustration in Figure 10 has a problem and then no A, B, C.
Answer: We do not understand the suggestion. However, the graph presented was revised and the information corresponds.
- The authors should expand the sample size of the data to make the antibacterial effect of MnO2more realistic, without accidental errors.
Answer: The effect of the concentration of the Nps will be evaluated in a future work, since the object of this article is the difference in behavior between bacteria and its estimation in suspension and deposited. But as the evaluator mentions, this specific analysis would render the explanation better.
- The authors should compare the antibacterial properties of MnO2with traditional antibiotics or other antibacterial organisms and carbon dots.
Answer: For this study, the comparison with traditional antibiotics or carbon points was not carried out, because the mechanism is different. In these results, the formation of ROS via photocatalysis is evaluated, there are antibiotics that undergo photolysis by themselves. However, some under light effects can generate singlet oxygen and that would be a way to buy. In the case of carbon dots, we must favor confinement in these nanoparticles so as not to incur dimensionality effects. But for later publications of course we will consider it.
- Reference format needs to be carefully checked and corrected.
Answer: All the references were corrected, as suggest the evaluator.

Round 2
Reviewer 2 Report
Report on revised manuscript (Nanomaterials) – 1928601 R1
"Manganese dioxide nanoparticles prepared by laser ablation as materials with interesting electronic, electrochemical, and disinfecting properties in colloidal suspension and deposited on FTO"
by Jhonatan Corrales, Jorge Acosta-Vergara, Sandra Castro, Henry Riascos, Efraim Serna- Galvis, Ricardo Torres-Palma and Yenny Ávila- Torres
intended for publication in Nanomaterials
Summary
This manuscript intended for publication in Nanomaterials as a regular article provides an overview on new results of the production of MnO2-nanoparticles using laser ablation with various wavelengths. The nanoparticles were deposited by dip-coating on fluorine-doped tin oxide. Several physico-chemical characterizations were performed: optical and spectroscopic properties have been determined as well as electro-chemical studies were carried out. Key applications were disinfecting properties studied in colloidal and deposited regimes using E. coli and S. aureus microorganisms. Although the experiments were well described, some passages needed further detailed explanation which has been done in the revised version. Nevertheless, there is still a number of typos and some ambiguous descriptions in the text: please correct them carefully!
After correction, I recommend this work for publication in Nanomaterials.
List of remarks and corrections
· abstract, page 1, line 31: band gap (typo)
· introduction, page 2, line 72: …NPs have… (typo)
· introduction, page 3, line 93: …prepared by… (please insert a blank)
· page 4, line 167: please clarify the unit sq-1 (this seems to be no SI unit)
· page 6, line 234: significantly (typo)
· page 6, fig. 1: I would recommend to combine both graphs in a single one to emphasize the changes of absorbance when altering the ablation duration
· page 7, line 269: …reported in the literature…Here, the respective reference(s) should be mentioned
· page 11, line 354: electrode (typo)
· page 13, line 433, please delete …the…(typo)
· page 14, line 456: …E. coli is higher than…(log Unit??) What do you mean by “log unit”? This unit does not exist! Please describe it in a way like factor 10 or “one order of magnitude”)
Author Response
Dear Evaluator
We appreciate all your corrections.
Our article has improved significantly. Below we detail the corrections.
- abstract, page 1, line 31: band gap (typo), introduction, page 2, line 72: …NPs have… (typo), introduction, page 3, line 93: …prepared by… (please insert a blank), page 6, line 234: significantly (typo):
Answer: All corrections were incorporated to manuscript.
- Pag 4, Line 167: please clarify the unit sq-1(this seems to be no SI unit).
Answer: We have converted to known units, in the international system as follow:
(2.33 ꭥ cm-1 on a glass substrate XOP Glass, 25mm x 25mm).
- page 6, fig. 1: I would recommend to combine both graphs in a single one to emphasize the changes of absorbance when altering the ablation duration.
Answer: We combine both graphs in a single one, as follow:
Figure 1. UV-vis spectra of NPs in colloidal suspension obtained at diverse laser ablation energies and 5 min and 10 min of ablation.
- page 7, line 269: …reported in the literature…Here, the respective reference(s) should be mentioned
Answer: The reference was included in the manuscript.
45). M. Selvam, S.R. Srither, S.R. Saminathan, “Chemically and electrochemically prepared graphene/MnO2 nanocomposite electrodes for zinc primary cells: a comparative study”, Ionics, pp. 791- 799, 2015, Dec 2017. doi.org/10.1007/s11581-014-1234-9.
- page 11, line 354: electrode (typo), page 13, line 433, please delete …the…(typo).
Answer: All corrections were incorporated to manuscript.
- coli is higher than…(log Unit??) What do you mean by “log unit”? This unit does not exist! Please describe it in a way like factor 10 or “one order of magnitude”)
Answer: We have corrected as suggested by the evaluator, as follow: it can also be noted that in both systems the inactivating action on E. coli is higher than on S. aureus (one order of magnitude).
Reviewer 3 Report
The author answered all the questions and I agree to accept the manuscript.
Author Response
We re-submit hereby the manuscript entitled: Manganese dioxide nanoparticles prepared by laser ablation as materials with interesting electronic, electrochemical, and disinfecting properties in colloidal suspension and deposited on FTO.
Thank you very much to the Editor and reviewers for the valuable corrections/suggestions made to our manuscript, which have been very useful to improve the quality of the document. Below you can find the authors answer to each one of the addressed issues.
Reviewer 2:
"Manganese dioxide nanoparticles prepared by laser ablation as materials with interesting electronic, electrochemical, and disinfecting properties in colloidal suspension and deposited on FTO". by Jhonatan Corrales, Jorge Acosta-Vergara, Sandra Castro, Henry Riascos, Efraim Serna- Galvis, Ricardo Torres-Palma and Yenny Ávila- Torres, intended for publication in Nanomaterials
Summary
This manuscript intended for publication in Nanomaterials as a regular article provides an overview on new results of the production of MnO2-nanoparticles using laser ablation with various wavelengths. The nanoparticles were deposited by dip-coating on fluorine-doped tin oxide. Several physico-chemical characterizations were performed: optical and spectroscopic properties have been determined as well as electro-chemical studies were carried out. Key applications were disinfecting properties studied in colloidal and deposited regimes using E. coli and S. aureus microorganisms. Although the experiments were well described, some passages needed further detailed explanation which has been done in the revised version. Nevertheless, there is still a number of typos and some ambiguous descriptions in the text: please correct them carefully!
After correction, I recommend this work for publication in Nanomaterials.
- abstract, page 1, line 31: band gap (typo), introduction, page 2, line 72: …NPs have… (typo), introduction, page 3, line 93: …prepared by… (please insert a blank), page 6, line 234: significantly (typo):
Answer: All corrections were incorporated to manuscript.
- Pag 4, Line 167: please clarify the unit sq-1(this seems to be no SI unit).
Answer: We have converted to known units, in the international system as follow:
(2.33 ꭥ cm-1 on a glass substrate XOP Glass, 25mm x 25mm).
- page 6, fig. 1: I would recommend to combine both graphs in a single one to emphasize the changes of absorbance when altering the ablation duration.
Answer: We combine both graphs in a single one, as follow:
Figure 1. UV-vis spectra of NPs in colloidal suspension obtained at diverse laser ablation energies and 5 min and 10 min of ablation.
- page 7, line 269: …reported in the literature…Here, the respective reference(s) should be mentioned
Answer: The reference was included in the manuscript.
45). M. Selvam, S.R. Srither, S.R. Saminathan, “Chemically and electrochemically prepared graphene/MnO2 nanocomposite electrodes for zinc primary cells: a comparative study”, Ionics, pp. 791- 799, 2015, Dec 2017. doi.org/10.1007/s11581-014-1234-9.
- page 11, line 354: electrode (typo), page 13, line 433, please delete …the…(typo).
Answer: All corrections were incorporated to manuscript.
- coli is higher than…(log Unit??) What do you mean by “log unit”? This unit does not exist! Please describe it in a way like factor 10 or “one order of magnitude”)
Answer: We have corrected as suggested by the evaluator, as follow: it can also be noted that in both systems the inactivating action on E. coli is higher than on S. aureus (one order of magnitude).
